# Intensity-Based Prompt Generation for Multi-Modality 3D Medical Image Segmentation

Viraj Aher[1][0009−0004−3859−2766]*, Sebastián A. Cajas
Ordóñez[1][0000−0003−0579−6178]*, Laura García[2][0009−0004−0709−8028]*,
Eliana Salas Villa[3][0000−0001−7059−704X]*, Luis Torres[3][0009−0001−0706−0626]*,
and Vinay K. Verma[4][0009−0009−4650−9666]* †

[1] National Irish Centre for AI (CeADAR), University College Dublin, Dublin, Ireland
[2] NeuroCo Student Research Group, University of Antioquia, Medellín, Colombia
[3] Université Rennes, CLCC Eugène Marquis, Inserm, Rennes, France
[4] Department of Computer Science and Engineering, IIIT Delhi, Delhi, India
`vinayv@iiitd.ac.in`
*all authors contributed equally
†corresponding author

**Abstract.** Interactive 3D medical image segmentation methods typically require manual bounding box prompts, limiting their applicability in automated workflows. In this work, we propose an intensity-based thresholding strategy that automatically generates bounding box prompts when explicit annotations are unavailable. Our method leverages statistical properties of medical images to identify regions of interest through adaptive thresholding, morphological operations, and connected component analysis. Experiments on the CVPR BiomedSegFM dataset demonstrate that this automated prompting strategy significantly improves segmentation performance in high-contrast modalities, achieving 0.73 DSC for CT and 0.74 DSC for PET, compared to 0.68 and 0.59 respectively with standard prompts. However, the method faces challenges in low-contrast modalities such as Ultrasound, where performance decreases from 0.68 to 0.31 DSC due to speckle noise and ambiguous tissue boundaries. We also report preliminary experiments with lightweight MobileNet encoders as alternatives to Vision Transformers, finding that current lightweight architectures suffer substantial accuracy degradation in 3D medical segmentation tasks. Our results highlight both the promise and limitations of automated prompt generation for multi-modality medical imaging. Code: https://github.com/lexorcvpr/lexor-cvpr-2025

**Keywords:** Prompt-based Segmentation · 3D Medical Image Segmentation · Automated Prompt Generation · Multi-Modality Learning

## 1 Introduction

The rapid advancement of biomedical imaging technologies has generated increasingly complex 3D medical datasets spanning modalities such as CT, MRI,

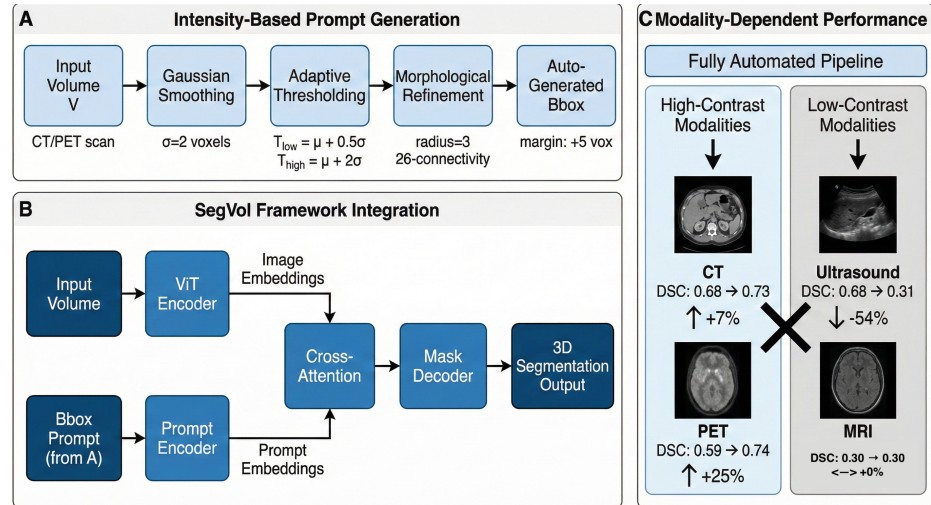

Fig. 1: Overview of the Intensity-Based Auto-Prompting Pipeline and Performance Analysis.

PET, and Ultrasound [2]. Deep learning approaches, from Fully Convolutional Networks [15] and U-Net [23] to self-configuring frameworks like nnU-Net [11], have achieved remarkable performance in medical image segmentation. Large-scale benchmarks including the Medical Segmentation Decathlon [26], AbdomenCT-1K [21], FLARE [20,10], KiTS [7,8], LiTS [1], autoPET [5,4], and TotalSegmentator [28] have accelerated progress across anatomical structures and imaging protocols.

Recent breakthroughs in 2D interactive segmentation, exemplified by the Segment Anything Model (SAM) [14] and SAM2 [22], have transformed how users interact with segmentation systems through point clicks and bounding boxes. Extensions to medical imaging include MedSAM [17] and MedSAM2 [19], which adapt these foundations to clinical applications. Several efficient variants have emerged targeting deployment constraints: MobileSAM [31] employs knowledge distillation for mobile deployment, EfficientViT-SAM [32] maintains accuracy while reducing computational cost, and FastSAM3D [25] accelerates volumetric processing.

The translation to 3D medical imaging has produced specialized architectures. SAM-Med3D [27] extends SAM to volumetric data, SegVol [3] introduces universal prompting for diverse anatomical targets, and VISTA3D [6] provides a unified foundation for CT segmentation. Interactive refinement approaches such as nnInteractive [12] and ProtoSAM-3D [24] incorporate iterative user feedback. Text-guided models including BioMedParse [33], CAT [9], and SAT [34] leverage anatomical priors through natural language, though they remain constrained by modality-specific training requirements.

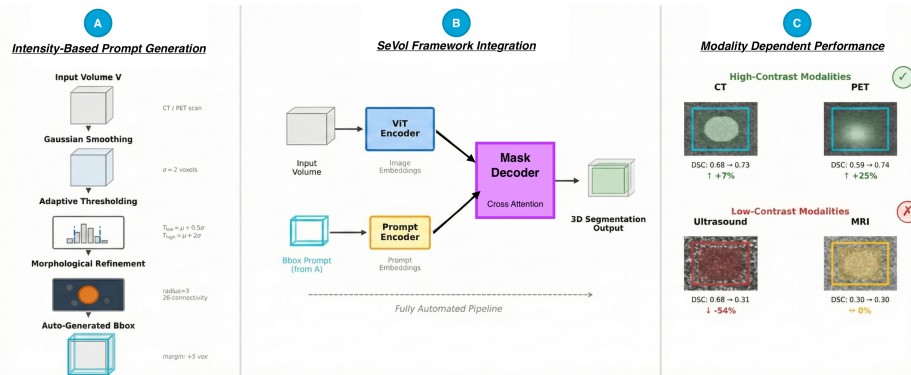

Fig. 2: Overview of our intensity-based automatic prompt generation for 3D medical image segmentation.

**(A) Prompt Generation:** Input volumes undergo Gaussian smoothing ($\sigma$=2), adaptive intensity thresholding ($T_{low} = \mu + 0.5\sigma_I$, $T_{high} = \mu + 2\sigma_I$), morphological refinement (radius 3), and bounding-box extraction (+5 voxel margin).

**(B) SegVol Integration:** The generated boxes replace manual prompts; the ViT encoder processes the volume while the prompt encoder processes the box, with cross-attention fusion yielding the final 3D mask.

**(C) Modality Effects:** The method boosts performance in high-contrast modalities (CT: 0.68→0.73 DSC; PET: 0.59→0.74), drops in low-contrast ultrasound (0.68→0.31), and remains unchanged for MRI (0.30).

Despite these advances, a fundamental bottleneck persists: these models rely on user-provided prompts for each segmentation. This creates scalability limitations in clinical workflows where hundreds of volumes require processing. The challenge intensifies with weakly annotated datasets lacking comprehensive bounding box labels.

In this work, we address this bottleneck by proposing an intensity-based method for automatic bounding box generation. Our approach exploits a core property of medical imaging: in many modalities, anatomical structures exhibit distinct intensity characteristics relative to surrounding tissues. By applying adaptive thresholding based on image statistics, we generate pseudo-prompts that approximate manual annotations without human intervention.

We evaluate our method within the SegVol framework on the CVPR Biomed-SegFM dataset [18]. Our experiments reveal a clear pattern: the intensity-based strategy excels in high-contrast modalities where tissue boundaries correlate with intensity gradients (CT, PET), but struggles where this assumption fails (Ultrasound, MRI), as illustrated in Figure 2.

Additionally, we report preliminary experiments with lightweight MobileNet encoders as alternatives to Vision Transformers. While model efficiency is desirable for clinical deployment, our results indicate substantial accuracy degra-

dation with current lightweight architectures, suggesting further innovation is needed before efficient encoders can match transformer performance.

Our contributions are:

1. An intensity-based automatic prompt generation method that eliminates manual annotation requirements during inference, enabling fully automated segmentation pipelines.
2. A modular encoder evaluation framework with corrected model factory implementation, enabling systematic comparison of different backbone architectures within the SegVol pipeline.
3. Systematic evaluation across four imaging modalities, characterizing when automated prompts succeed (CT: $+7\%$, PET: $+25\%$) and when they fail (Ultrasound: $-54\%$, MRI: no change).
4. Preliminary negative results on lightweight encoders, quantifying the substantial accuracy gap (34% relative decrease) between Vision Transformers and MobileNet-based architectures for 3D medical segmentation.

## 2   Method

Our approach builds upon the SegVol framework [3], which combines an image encoder, prompt encoder, and mask decoder for interactive volumetric segmentation. We retain the original architecture while introducing two key modifications: (1) an automated prompt generation module that operates during inference, and (2) a corrected model factory that enables systematic evaluation of alternative encoder architectures. Figure 2 provides an overview of our complete pipeline.

### 2.1   Framework Overview

The segmentation pipeline consists of three main components, as shown in Figure 2B. The image encoder processes 3D medical volumes to extract feature representations. We use a Vision Transformer (ViT) backbone following the original SegVol design, which employs self-attention mechanisms to capture long-range spatial dependencies in volumetric data. The prompt encoder transforms spatial inputs (bounding boxes or points) into embeddings that condition the segmentation. The mask decoder, adapted from SAM [14], fuses image and prompt features through cross-attention to produce the final segmentation mask.

### 2.2   Prompt Encoding

The prompt encoder supports two types of spatial prompts:

**Bounding Box Prompts.** Box coordinates $(x_1, y_1, z_1, x_2, y_2, z_2)$ defining the corners of a 3D region are normalized relative to volume dimensions and encoded using sinusoidal positional encoding to preserve spatial relationships.

**Point Prompts.** Positive points (indicating foreground) and negative points (indicating background) are processed through learnable embeddings. The encoder maintains four learnable embeddings: two for positive points and two for negative points, enabling the model to distinguish click types.

During training, we simulate user interaction by generating random prompts around ground truth regions. Bounding boxes are perturbed with $\pm 10\%$ spatial jitter, and points are sampled randomly within (positive) or outside (negative) target regions. We apply 20% prompt dropout (randomly omitting one prompt type during training) to encourage robustness when prompts are sparse or incomplete.

### 2.3   Intensity-Based Automatic Prompt Generation

Our key contribution is an automatic method for generating bounding box prompts when manual annotations are unavailable, as detailed in Figure 2A. The algorithm exploits statistical intensity properties common to many medical imaging modalities.

**Preprocessing.** Given a 3D volume $V$ normalized to the range $[0, 255]$, we first apply Gaussian smoothing with standard deviation $\sigma = 2$ voxels. This step reduces high-frequency noise and imaging artifacts that could fragment the subsequent thresholding, particularly important for modalities like Ultrasound that exhibit speckle patterns. The smoothing operation is implemented using a 3D Gaussian kernel applied via separable convolution for computational efficiency.

**Adaptive Thresholding.** Rather than using fixed intensity thresholds that would fail across diverse modalities and acquisition protocols, we compute adaptive thresholds based on the volume's statistical properties:

$$T_{low} = \mu + 0.5\sigma_I, \quad T_{high} = \mu + 2.0\sigma_I \tag{1}$$

where $\mu$ and $\sigma_I$ denote the mean and standard deviation of voxel intensities in the smoothed volume. Voxels with intensities falling between $T_{low}$ and $T_{high}$ are selected as candidate foreground regions. This formulation targets structures that are statistically brighter than average (appropriate for contrast-enhanced regions in CT, metabolically active regions in PET, and hyperintense lesions) while excluding very bright outliers that often correspond to artifacts.

**Morphological Refinement.** The initial binary mask typically contains holes and fragmented regions. We apply morphological closing using a 3D spherical structuring element with radius $r = 3$ voxels. Closing (dilation followed by erosion) fills small gaps within structures while preserving the overall boundaries. This step connects nearby voxels that may have been separated by noise or partial volume effects.

**Connected Component Analysis.** After morphological refinement, we perform connected component labeling on the binary mask using 26-connectivity (considering all adjacent voxels in 3D). Multiple distinct regions may satisfy our intensity criteria. We select the largest connected component by voxel count as the primary structure of interest, based on the assumption that the target anatomy constitutes the dominant high-intensity region in the volume. For multi-lesion cases, this heuristic selects the largest lesion, which may not always match clinical priorities.

**Bounding Box Extraction.** From the selected component, we compute the axis-aligned 3D bounding box as the minimum enclosing cuboid. A margin

---

**Algorithm 1** Intensity-Based Bounding Box Generation

---

**Require:** 3D volume $V$
**Ensure:** Bounding box coordinates $\mathcal{B}$
    **Parameters:** $\sigma \leftarrow 2$, $r \leftarrow 3$, $m \leftarrow 5$
 1: $V_s \leftarrow \textsc{GaussianSmooth}(V, \sigma)$                       ▷ Reduce noise
 2: $\mu \leftarrow \textsc{Mean}(V_s)$
 3: $\sigma_I \leftarrow \textsc{Std}(V_s)$
 4: $T_{\text{low}} \leftarrow \mu + 0.5\sigma_I$
 5: $T_{\text{high}} \leftarrow \mu + 2.0\sigma_I$
 6: $M \leftarrow (T_{\text{low}} < V_s < T_{\text{high}})$          ▷ Generate binary mask
 7: $M \leftarrow \textsc{MorphClose}(M, r)$        ▷ Fill morphological holes
 8: $\mathcal{C} \leftarrow \textsc{ConnectedComponents}(M)$
 9: $C_{\text{max}} \leftarrow \arg\max_{C \in \mathcal{C}} |C|$         ▷ Select largest component
10: $(x_1, y_1, z_1), (x_2, y_2, z_2) \leftarrow \textsc{GetBounds}(C_{\text{max}})$
11: $\mathcal{B} \leftarrow (x_1 - m, y_1 - m, z_1 - m, x_2 + m, y_2 + m, z_2 + m)$
12: **return** $\mathcal{B}$

---

of $m = 5$ voxels is added to each side of the box to ensure complete coverage of boundary voxels and provide spatial context for the segmentation model.

Algorithm 1 provides the complete procedure.

**Modality Considerations.** The threshold parameters were selected empirically on a validation subset and remained fixed across all experiments. For CT images, where Hounsfield units provide absolute density measurements, the method reliably identifies high-density structures (bone, contrast-enhanced vessels) or isolates soft tissues from surrounding fat and air. For PET images, regions of elevated metabolic uptake (high standardized uptake values) naturally exceed the intensity thresholds. However, in MRI and Ultrasound, where intensity values are relative and heavily dependent on acquisition parameters (flip angle, echo time) or corrupted by speckle noise, the method produces less reliable prompts. Figure 2C illustrates these modality-dependent outcomes.

### 2.4 Decoder Architecture

We retain the SAM mask decoder architecture [14] with modifications for 3D processing. The decoder consists of a two-layer transformer module with self-attention and cross-attention mechanisms. Cross-attention enables bidirectional information flow between image embeddings and prompt embeddings, spatially aligning features with the prompted region.

Following established practice in medical image segmentation [16], we train the model with a combined loss function:

$$\mathcal{L} = \lambda_1 \mathcal{L}_{Dice} + \lambda_2 \mathcal{L}_{Focal} \tag{2}$$

where $\mathcal{L}_{Dice}$ addresses class imbalance through overlap maximization and $\mathcal{L}_{Focal}$ emphasizes hard examples near boundaries. We set $\lambda_1 = \lambda_2 = 1.0$.

## 2.5   Model Factory and Encoder Evaluation

To systematically evaluate alternative encoder architectures, we developed a corrected model factory that properly instantiates and integrates different backbone networks within the SegVol pipeline. The original codebase contained implementation issues that prevented fair comparison across encoders. Our corrected factory ensures consistent initialization, proper weight loading, and correct feature map dimensions for each encoder variant.

We tested MobileNet-based architectures adapted for 3D inputs using depthwise separable convolutions, inspired by efficient 2D segmentation approaches [31,32]. The MobileNet3D encoder follows a hierarchical design with progressive channel expansion ($32{\rightarrow}64{\rightarrow}128{\rightarrow}256{\rightarrow}512$) using depthwise separable blocks. Each block applies a depthwise 3D convolution followed by a pointwise ($1{\times}1{\times}1$) convolution, reducing computational cost compared to standard convolutions.

We emphasize that these lightweight encoder experiments represent preliminary negative results: the accuracy degradation was substantial, indicating that further research is needed before lightweight encoders can be practical for 3D medical segmentation.

## 3   Experiments

### 3.1   Dataset

We evaluate on the CVPR BiomedSegFM challenge dataset [18], an extension of the CVPR 2024 MedSAM on Laptop Challenge. The dataset aggregates 3D volumes from multiple public sources [26,7,1,21,5] spanning Computed Tomography (CT), Magnetic Resonance Imaging (MRI), Positron Emission Tomography (PET), and Ultrasound (US). Annotations include organ and lesion segmentations with corresponding bounding boxes, created using 3D Slicer [13], ITK-SNAP [30], and MedSAM2 [19].

We use the coreset track, which provides 10% of the full training data for model development. This constraint tests generalization under limited supervision, a realistic scenario for many clinical applications where annotated data is scarce.

### 3.2   Evaluation Metrics

Performance is measured using the Dice Similarity Coefficient (DSC):

$$DSC = \frac{2|P \cap G|}{|P| + |G|} \tag{3}$$

where $P$ is the predicted segmentation and $G$ is the ground truth. DSC ranges from 0 (no overlap) to 1 (perfect agreement). We report average DSC across all test cases as well as per-modality breakdowns.

Table 1: Segmentation performance (DSC) comparing manual prompts versus automatic intensity-based prompts. The intensity method improves CT and PET segmentation but degrades Ultrasound performance.

| Prompt Method | Mean | CT | MRI | PET | US |
|---|---|---|---|---|---|
| Manual (Baseline) | **0.50** | 0.68 | 0.30 | 0.59 | **0.68** |
| Intensity-Based (Ours) | 0.47 | **0.73** | 0.30 | **0.74** | 0.31 |

### 3.3  Implementation Details

**Preprocessing.** Following the MedSAM protocol [17], all volumes were normalized to the intensity range [0, 255]. For CT, we applied clinical windowing: soft tissue (W: 400, L: 40), lung (W: 1500, L: $-160$), brain (W: 80, L: 40), and bone (W: 1800, L: 400). For MRI, PET, and Ultrasound, intensities were clipped to the 0.5th and 99.5th percentiles before linear normalization to reduce outlier effects.

**Training Configuration.** Models were trained using the Adam optimizer with learning rate $1 \times 10^{-5}$. Training proceeded for up to 3000 epochs with learning rate halving every 200 epochs. Experiments were conducted on two NVIDIA H100 80GB GPUs with approximately 3 hours total training time.

**Environment.** Red Hat Enterprise Linux 8.6, Python 3.11, PyTorch 2.0, CUDA 12.5.

**Model Specifications.** The ViT encoder contains approximately 86M parameters. Our MobileNet variants range from 4M to 10M parameters. Computational cost is approximately 22.1 GFLOPs per inference.

## 4  Results

### 4.1  Impact of Intensity-Based Prompts

Table 1 compares segmentation performance with standard manual prompts versus our automatic intensity-based prompts. Both configurations use the same ViT encoder; only the prompt source differs. The modality-dependent pattern is visually summarized in Figure 2C.

The results reveal a modality-dependent pattern:

**High-Contrast Modalities (CT, PET).** The intensity-based method substantially improves performance. CT segmentation increases from 0.68 to 0.73 DSC ($+7.4\%$), and PET improves dramatically from 0.59 to 0.74 DSC ($+25.4\%$). In these modalities, structures of interest have distinct intensity profiles: bones and organs in CT have characteristic Hounsfield units, while metabolically active regions in PET show elevated tracer uptake. Our adaptive thresholding successfully identifies these regions, generating accurate bounding boxes that guide segmentation.

Table 2: Comparison of encoder architectures using our corrected model factory. Lightweight encoders show substantial accuracy degradation compared to the ViT baseline. These results represent preliminary negative findings.

| Encoder | Parameters | Avg DSC |
|---|---|---|
| ViT (Baseline) | ∼86M | **0.50** |
| ViT + Intensity Prompts | ∼86M | 0.47 |
| MobileNetV3 (1000 epochs) | ∼5.4M | 0.33 |
| MobileNet (100 epochs) | ∼4.0M | 0.25 |
| FastViT (100 epochs) | ∼10M | 0.24 |

**Low-Contrast Modalities (Ultrasound).** Performance drops sharply from 0.68 to 0.31 DSC ($-54.4\%$). Ultrasound images suffer from speckle noise, acoustic shadows, and intensity variations unrelated to tissue boundaries. The thresholding algorithm produces unreliable boxes that often exclude the target structure or include excessive background, degrading segmentation quality.

**Variable-Contrast Modalities (MRI).** Performance remains unchanged at 0.30 DSC. MRI intensity depends on pulse sequence parameters (T1, T2, FLAIR) rather than absolute tissue properties, making statistical thresholding inconsistent. While not harmful, the intensity method provides no benefit over manual prompts for MRI.

### 4.2   Lightweight Encoder Experiments

Table 2 reports results from our corrected model factory comparing the ViT encoder against lightweight alternatives.

The lightweight encoders achieve significantly lower accuracy despite extended training. MobileNetV3, with 94% fewer parameters than ViT, reaches only 0.33 DSC after 1000 epochs. The accuracy gap of 0.17 DSC (34% relative decrease) indicates that current lightweight architectures lack sufficient representational capacity for multi-modal 3D medical segmentation.

We note several limitations of these experiments: (1) the lightweight encoders were trained from scratch rather than with pretraining, (2) training was conducted on the 10% coreset which may disadvantage data-hungry architectures, and (3) architectural adaptations for 3D processing may not have been optimal. These results should be interpreted as preliminary negative findings rather than definitive conclusions about lightweight encoder viability.

### 4.3   Qualitative Analysis

Figure 3 shows representative CT segmentation results. The model accurately delineates anatomical structures with sharp boundaries, reflecting the high quantitative scores.

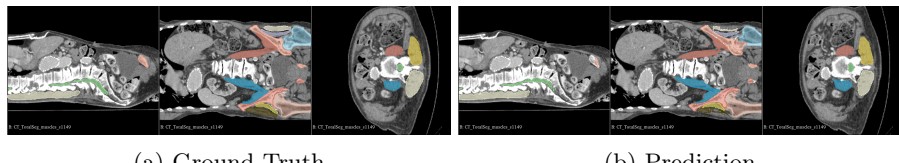

(a) Ground Truth                    (b) Prediction

Fig. 3: Qualitative results on abdominal CT. The model successfully segments spine and muscle groups. The distinct Hounsfield units of these tissues enable accurate automatic prompt generation.

Figure 4 illustrates a failure case in brain MRI tumor segmentation. The prediction exhibits boundary leakage, where the segmentation extends beyond the tumor into surrounding edema.

These qualitative examples confirm the quantitative findings: our method excels when intensity correlates with anatomical boundaries (CT) but struggles when this assumption fails (MRI).

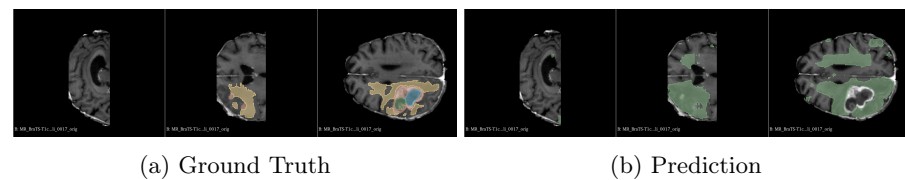

(a) Ground Truth                    (b) Prediction

Fig. 4: Failure case on brain MRI (T1c). The prediction shows boundary leakage into surrounding tissue. MRI's relative intensity scale makes automatic prompt generation unreliable, and the model struggles to distinguish tumor margins from edema.

## 5   Discussion

### 5.1   When Does Intensity-Based Prompting Work?

Our results identify a clear criterion for automatic prompt success: the method works when target structures have statistically distinct intensity distributions from surrounding tissues. This condition holds in modalities with absolute intensity scales. In CT, Hounsfield units directly measure tissue density, enabling reliable differentiation of bone, soft tissue, fat, and air. In PET, standardized uptake values quantify metabolic activity, making hypermetabolic lesions statistically salient.

The condition fails in modalities with relative intensity scales. In MRI, signal intensity depends on acquisition parameters (flip angle, repetition time, echo time) and scanner calibration rather than intrinsic tissue properties. The same

tissue can appear bright or dark depending on the pulse sequence. In Ultrasound, speckle noise dominates the intensity distribution, and acoustic impedance variations create artifacts unrelated to anatomical boundaries.

This finding has practical implications. Automated pipelines for CT or PET analysis can leverage intensity-based prompts to eliminate manual annotation bottlenecks. For MRI or Ultrasound, alternative prompting strategies are needed, such as atlas-based initialization, learned prompt prediction networks, or hybrid approaches combining intensity cues with anatomical priors.

## 5.2 The Efficiency Gap in 3D Medical Segmentation

Our lightweight encoder experiments, enabled by the corrected model factory, reveal a substantial accuracy gap between Vision Transformers and efficient CNNs. The 34% relative performance drop when moving from ViT to MobileNet suggests that 3D medical segmentation requires architectural capacity that current lightweight designs cannot provide.

This contrasts with 2D scenarios where MobileSAM [31] and EfficientViT-SAM [32] achieve competitive performance with dramatically reduced computation. The difference may stem from the increased complexity of 3D spatial relationships, where attention mechanisms in ViT capture long-range dependencies more effectively than local convolutions.

This negative result is valuable for the field. It indicates that naive model compression through encoder substitution is insufficient; more sophisticated approaches such as knowledge distillation from 3D transformers, neural architecture search optimized for medical imaging, or hybrid transformer-CNN designs may be necessary to achieve both efficiency and accuracy.

## 5.3 Limitations

Our study has several limitations. First, the intensity-based method uses fixed threshold parameters ($T_{low} = \mu + 0.5\sigma$, $T_{high} = \mu + 2.0\sigma$) that may not generalize optimally across institutions, scanner vendors, or reconstruction algorithms. Adaptive parameter selection based on modality metadata or learned calibration could improve robustness.

Second, our evaluation is limited to the coreset track (10% of training data), which may not reflect performance with full training data. The lightweight encoder gap might narrow with more training examples.

Third, the intensity method assumes a single dominant structure of interest (selecting the largest connected component). Multi-lesion scenarios or cases with multiple anatomical targets would require modified selection criteria.

Fourth, the lightweight encoder experiments were preliminary and did not explore pretraining strategies, knowledge distillation from larger models, or architectural optimizations specifically designed for 3D medical data.

## 6    Conclusion

We presented an intensity-based method for automatic bounding box prompt generation in 3D medical image segmentation. Our approach eliminates the need for manual prompts during inference by leveraging statistical intensity properties to identify regions of interest. Experiments on the CVPR BiomedSegFM dataset demonstrate that this strategy significantly improves segmentation in high-contrast modalities (CT: $+7.4\%$, PET: $+25.4\%$) while facing challenges in low-contrast modalities (Ultrasound: $-54.4\%$).

We also contributed a corrected model factory enabling systematic encoder comparison, and reported preliminary negative results on lightweight encoder alternatives, finding that MobileNet-based architectures suffer substantial accuracy degradation (34% relative decrease) compared to Vision Transformers. This highlights an open challenge: developing efficient encoders suitable for 3D medical imaging without sacrificing segmentation quality.

Future work should explore learned prompt prediction networks that can adapt to modality-specific characteristics, as well as advanced model compression techniques such as knowledge distillation to bridge the efficiency gap for clinical deployment.

**Acknowledgements**  We thank all the data owners for making the medical images publicly available and CodaLab [29] for hosting the challenge platform.

**Disclosure of Interests.**  The authors have no competing interests to declare.

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
