# OpenReview forum: "Intensity-Based Prompt Generation for Multi-Modality 3D Medical Image Segmentation"
_thecvf.com/CVPR/2025/Workshop/MedSegFM — CVPR 2025 Workshop MedSegFM Submission_

### Official Review · Reviewer_7PL6 · 2025-09-08
**MobileSeg3D: A Lightweight Framework for Multi-Modality 3D Medical Image Segmentation**

**Rating:** 7
**Confidence:** 4

**Review:**

This work proposes a lightweight, multi-modal 3D medical image segmentation framework that demonstrates considerable innovation and potential for application. The experimental design is sound, but there is room for improvement in method comparison and failure case analysis. The authors are advised to address the following comments and make appropriate revisions：

1、Although the method of generating bounding boxes based on intensity thresholds is practical, it relies on image contrast and performs poorly in low-contrast modalities.

2、It performed well on CT and PET (DSC > 0.70), but poorly on MRI and US(DSC ~0.30). The reasons for this need to be thoroughly analyzed and directions for improvement proposed.

3、The presentation of qualitative results is limited, especially the analysis of failure cases is insufficient. It is recommended to increase the visualization of typical error cases and discuss improvement strategies.

4、The work has a clear structure and fluent language, but some method details (such as the specific parameters and training strategies of MobileNet3D) are not described in detail enough, which is not conducive to reproduction.

5、The clarity of Figure 1 is poor and the font is small.

I hope the author will revise this work in light of the above issues.

---

### Official Review · Reviewer_4qRd · 2025-09-14
**Comments**

**Rating:** 6
**Confidence:** 4

**Review:**

The paper presents an approach that leads CVPR challenge and achieves competitive segmentation accuracy while remaining computationally efficient. Several points I would like to be addressed by the authors:

1. When the authors talk about `modular segmentation framework`in the abstract, I am not sure why the proposed approach should be considered as a modular method. How many modules do you have?

2. The arguement on the using MobileNet3D computational efficiency can be clearer. I would like an illustration on how the MobileNet3D makes the visual encoder linear computational complexity.

3. The part on the `intensity-based prompt generation` is not well illustrated. Please give a demonstration on what do you mean by `weak` or `sparse` setting and how your proposal works to fix the issue.

4. I think it is good to talk about inference speed. Yet what do you do to achieve the efficiency? Is that the usage of lightweight encoder ensures the speed or you have other strategies for the purpose? What is the criteria for the evaluation on the efficiency?

Please address the comments.

---

### Official Review · Reviewer_VsWD · 2025-09-15
**Review of "MobileSeg3D: A Lightweight Framework for Multi-Modality 3D Medical Image Segmentation"**

**Rating:** 6
**Confidence:** 4

**Review:**

###  Summary:

The authors propose a lightweight variation of SegVol for the segmentation of 3D medical images by replacing the computationally expensive 3D Vision Transformer image encoder with a MobileNet3D architecture.

### Strengths:

* Slightly improved segmentation performance combined with faster inference speed.
* Thorough evaluation and ablation of various architectures for the image encoder. Skip connections were also tested, but they had a negative impact on performance.

### Weaknesses:

* The intensity-based prompt generation approach is insufficiently described. It remains unclear whether a point prompt is used as the basis, and there is no quantitative evaluation of its impact on performance.

### Clarity:

The paper is well written and easy to follow.

### Suggestions:

1. The code link in the abstract should be clickable.
2. Figure 1 should be rendered as a vector graphic (e.g., SVG) to improve readability, as the current version appears blurry.
3. In Section 2.2, the subsection “Training Simulation” should be rewritten as either complete sentences or structured bullet points for improved clarity.
4. Section 2.4, titled “Post-processing,” describes an intensity-based strategy for bounding box prompt generation. However, this appears to be a pre-processing step and would be more appropriately placed in Section 3.2 (“Preprocessing”).
5. References (6) and (7) in Table 2 for “Number of model parameters” and “Number of FLOPs” are non-functional and should be corrected.
6. The mention of text prompts via a frozen CLIP encoder in Section 2 is not substantiated in the remainder of the paper and should be removed unless further explanation and evaluation are provided.

---

### Official Review · Reviewer_oiif · 2025-09-16
**All in all the paper seems marginally above acceptance treshold, it presents an efficiency improvement over the SegVol baseline but misses a deeper comparison to other baselines in terms of efficiency, it also could go into more detail about specifics of the approach itself.**

**Rating:** 6
**Confidence:** 4

**Review:**

The main claim of the paper is that replacing the 3D Vision Transformer backbone of SegVol by a MobileNet3D-based backbone improves efficiency without compromising on performance.
The authors also implemented an intensity-based prompt-generation method to support generating bounding-box prompts for datapoints with missing annotations.
Since the efficiency improvement and inference speed optimization seem to be claimed contributions I would have liked to see a more in-depth analysis about this.

All in all the paper seems marginally above acceptance treshold, it presents an efficiency improvement over the SegVol baseline but misses a deeper comparison to other baselines in terms of efficiency, it also could go into more detail about specifics of the approach itself.

### Presentation/Clarity
You claim that you optimized inference speed by implementing and benchmarking a fast inference pipeline, it is not clear to me what you did to increase the speed other than changing the backbone of the architecture. I am also missing a detailed benchmark regarding the inference speed, you mention 387ms and 2.3x speedup in Fig. 1, but there are no details on how exactly this was measured and also no comparison with the other baseline architectures from Table 3. A plot of the pareto front of inference time/predictive performance would have been nice, as well as a comparison to the other baselines provided in the competition.
I would also be interested in the quantitative results on the microscopy modality.
Mentioning "MobileNet-2.5D (initialized from scratch)" in the "Pre-trained Model" row in Table 2 is a bit confusing, maybe call it sth. like backbone architecture instead.
In Table 3 maybe rename "Validation Score (10% FM + Intensity prediction)" to "our method", the name validation score does not make sense.
Regarding Section 2.4 it is not clear to me whether this is something that you do during training or inference, also it seems like this is pre-processing not post-processing? Some more details on how exactly the mean and standard deviation are getting used here would have been helpful too.

### Reproducibility
The authors released their code on GitHub, the repository contains a python environment and the paper contains additional information about the environment and hardware the model was trained on.

### Other comments
- you mention that nnInteractive/ProtoSAM-3D require finetuning, what do you mean by that?
- "In both clinical and research workflows, the ability to generalize across modalities is increasingly critical, yet remains largely unsolved." - could you elaborate why this ability is critical in the clinical workflows?
- you sometimes refer to your backbone as MobileNet3D, but some other times you refer to it as MobileNet-2.5D?
- In the conclusion you mention "our method achieves an average DSC of 0.50 and ranks 4th on the official validation leaderboard, showcasing the effect of our method in both generalization and runtime efficiency.", I think the leaderboard and does not tell you anything about runtime efficiency (Also I think that the inference runtime was originally going to be a metric in the competition, but this was changed later on iirc)
- What does "Prompt dropout: 20" at the end of Section 2.2 mean?
- "Box Prompt Encoding" in Section 2.2 should be cursive for consistency
- The whole of Section 2.2 reads weirdly, it seems almost like some of the paragraphs are multiple bullet points of a list concatenated
- Was the training pipeline for all baseline architectures in Table 3 the same?
- In Section 2.3 you mention the output head "includes multiple mask prediction branches", what do you mean by that?

---

### Decision · Program_Chairs · 2025-11-12

Revision